# Use of a 3D Model with Reconstructed Human Epidermis Infected with Fungi and Covered with a Bovine Nail to Simulate Onychomycosis and to Evaluate the Effects of Antifungal Agents

**DOI:** 10.3390/jof11040285

**Published:** 2025-04-04

**Authors:** Francesca Giulia Urso, Salvatore Del Prete, Christelle Foucher, Martina Barberis, Francesco Carriero, Amandine Bart

**Affiliations:** 1The Novo Nordisk Foundation Center for Stem Cell Medicine, reNEW, University of Copenhagen, Norre Alle 14, 2200 København N, Denmark; francesca.giulia.urso@sund.ku.dk; 2Service Biotech, Via Croce Rossa, 23, 80131 Napoli, Italy; saldelp@gmail.com; 3Urgo Research Innovation & Development, 42 Rue de Longvic, 21300 Chenôve, France; a.bart@fr.urgo.com; 4Vitroscreen, Via Mosè Bianchi, 103, 20149 Milan, Italy; martina.barberis@vitroscreen.com (M.B.); francesco.carriero@vitroscreen.com (F.C.)

**Keywords:** onychomycosis, fungi, in vitro model, antifungal agents, *Trichophyton rubrum*, Scanning Electron Microscopy, reconstructed human epidermis

## Abstract

An in vitro 3D model using fungus colonized Reconstructed Human Epidermis (RHE) represents an effective preclinical model to simulate the pathological conditions of onychomycosis. We evaluated the suitability of this 3D onychomycosis model and use it to assess the effects of topical products on fungus growth and nail structure. Five sets of differentiated RHE were colonized with *Trichophyton rubrum* and covered with bovine nails. Colonized RHE with no product application (CNA) served as the control. Four different products classified as medical devices were applied once daily on the nails: Urgo Damaged Nails (UDN), Excilor (EXlor), Poderm Purifying (PDermP), Poderm Booster (PDermB). After 5 days, *T. rubrum* presence was visualized by the Grott Methenamine Silver staining method and quantified as the sum signal intensity of processed acquisitions. Fungal hyphae and the nail structure were analyzed by SEM. A semi-quantitative evaluation of fungal presence showed a reduction after UDN (−34%, *p* < 0.001) and EXlor (−28%, *p* < 0.020) applications compared to CNA. No significant difference was observed after PDermP applications (−2%). The nail structure appeared preserved after UDN applications and severely damaged after EXlor and PDermP applications. These findings demonstrate significant effects of different products on fungal growth and nail structure, suggesting that this 3D model might be a valuable tool for predicting the effects of antimycotic treatment in humans.

## 1. Introduction

Onychomycosis, a fungal infection of the toenail or fingernail bed, is a chronic nail disorder that represents one of the main reasons for primary care visits. Common signs and symptoms of onychomycosis include nails that appear discolored, deformed, hypertrophic, or hyperkeratotic [1]. In severe cases, the nail separates from the nail bed, gets brittle, and breaks. Onychomycosis can be painful, cause discomfort, and can negatively impact quality of life depending on the disease progression duration and gender [2]. The treatment of onychomycosis is highly challenging as this fungal infection is difficult to treat due to a high recurrence rate, which is related to the anatomic and pathophysiological characteristics of the nail and due to the development of fungal resistance [3,4]. Although long-term oral treatments for moderate to severe onychomycosis may be successful in achieving a complete mycological cure, they can cause serious adverse effects and are not suitable for use in elderly or immune-compromised individuals [5]. Poor adherence to long-term treatment may be an issue as well. Topical treatments of onychomycosis might be preferred as they do not induce systemic side effects or risks of drug interactions and may promote better patient compliance. One drawback is the potentially limited drug penetration into the keratin structure of the nail plate, which may hinder effective biofilm targeting [6,7,8].

Many in vitro models and ex-vivo models have been developed in the last decade to assess the penetration and effects of antifungal drugs topically applied to either human nails mounted in specific chambers seeded with *Trichophyton rubrum*, bovine hoof plates, or artificial nail plates such as keratin bio-membranes generated from human hair [9,10,11,12]. While most of them mainly evaluate the permeation capacity of antifungal agents, their inhibitory effects on fungus growth were assessed with models having fungi grown on the surface of the artificial nails and not beneath the nail, which is less indicative of the human onychomycotic nail. Moreover, most in vitro models do not reproduce the microenvironment of the nail bed, which provides viable conditions for fungal growth.

Thus, we aimed to develop a 3D model that provides appropriate conditions for fungal growth and simulates human onychomycosis. To best recreate an in vitro environment suitable for fungal infection, we selected a model that included (a) a 3D reconstructed human epidermis (RHE) in vitro, which represents an interesting test system due to its morphological similarity and metabolic activity to the skin, as well as being a fully viable substrate along with the nail; and (b) bovine nail sheets representing an essential substrate and microenvironment for the fungal growth and adhesion. This proof-of-concept study was designed to evaluate the suitability of such a 3D model as a preclinical onychomycosis model and to test the effects of different topical medical devices and non-traditional antifungal agents on fungal growth and nail structure.

## 2. Materials and Methods

### 2.1. Experimental Plan

The study was conducted on the following six arms in biological quadruplicate (except for the non-colonized RHE arm, which was in biological triplicate): (1) Non-Colonized (NC): sterile RHE tissue covered with a bovine nail with no product application, used as a negative control; (2) Colonized with no application (CNA): RHE colonized with *T. rubrum* covered with a bovine nail with no product application; (3) C/UDN: RHE colonized with *T. rubrum* covered with a bovine nail with one daily application of Urgo Damaged Nails; (4) C/PdermP: RHE colonized with *T. rubrum* covered with a bovine nail with one daily application of the natural oil-based Poderm Purifying; (5) C/PdermP + PdermB: RHE colonized with *T. rubrum* covered with a bovine nail with one daily application of Poderm Purifying + Poderm Booster; (6) C/EXlor: RHE colonized with *T. rubrum* covered with a bovine nail with one daily application of Excilor Forté. The Excilor product is the positive control as a similar effect of daily Excilor^®^ applications compared to once weekly amorolfine application was shown in vitro [13]. Additionally, clinically proven efficacy is being evaluated in subjects with onychomycosis [14]. All tested products are commercial products aiming at strengthening and restoring the nail structure damaged by onychomycosis. All products are classified as medical devices. The various steps in the set-up of the 3D model are summarized in Figure 1 and are detailed below.

### 2.2. Set-Up of the 3D Model of Colonized Reconstructed Human Epidermis Covered by a Bovine Nail

The SkinEthic^TM^ Reconstructed Human Epidermis (RHE, EPISKIN SA, Lyon, France) with an area of 0.5 cm^2^ was used to reproduce a differentiated epidermis formed after 10 days of air-lift culture of normal human keratinocytes from skin biopsies in a chemically defined medium. The RHE batch was tested for the absence of HIV, Hepatitis B, Hepatitis C, and Mycoplasma before use. Histological observation reveals a multi-layered (5.5 cell layers), differentiated epidermis consisting of organized basal, spinous, and granular layers and a multilayered stratum corneum (Figure 2b). The maintenance medium (SMM, EPISKIN SA, Lyon, France) was tested for sterility. The inserts containing the RHE at day 12 (Day 12 RHE) were placed at room temperature in a multiwell plate filled with an agarose nutrient solution in which they were embedded. Each RHE was cultivated in growth medium (SGM, EPISKIN SA, Lyon, France) for up to 16 days, corresponding to a full RHE differentiation (from D1 to D5) before being used for the study and fungal infection.

*Trichophyton rubrum* (*T. rubrum*, ATCC 28188, COMPLIFE, Lyon, France) culture was sown on Potato Dextrose Agar (Merck Life Science, Darmstadt, Germany—Agar 15 g/L—Dextrose 20 g/L-Potato Extract 4 g/L) to check the morphology. Another subculture was performed on Potato Dextrose Agar, starting from the fungus culture generated the previous week. It was incubated at 28 °C for 4–7 days under aerobic conditions to obtain a fresh culture. A viable count on agar plates was performed to check the starting concentration inoculum level of 10^6–7^ CFU/mL (OD_520nm_ = 0.5). The inoculum, defined by viable counts on Potato Dextrose Agar, was 1.1 × 10^7^ CFU/mL. On the day of RHE infection (D5), the fungal spores were picked from the second subculture using a sterile loop and by scratching the surface. Then, 30 μL of *T. rubrum* spores were used to colonize the entire tissue surface, except for the control (non-colonized RHE), which remained sterile. Infected RHE were incubated under optimized experimental conditions to permit fungus growth (30 °C 2% CO_2_ in a saturated humidity incubator).

After the fungus growth period, bovine nail sheets (XENOMETRIX AG, Allschwill, Switzerland) were carefully placed on the infected RHE apical surface within the tissue insert. Throughout the entire experimental phase, the infected RHE-nail system was incubated at 30 °C with 2% CO_2_ in a humidity-saturated incubator. Figure 2a shows the 3D model of the colonized RHE covered by a bovine nail. Figure 2b shows a histological observation of the healthy RHE, with the stratum corneum and the different epidermis layers being visible.

### 2.3. Application of Tested Products and Tissue Collection

Starting on D6, the test products were applied once daily directly on the bovine nails within the 5-day experimental window. Except for the two controls, non-colonized RHE and colonized RHE with no application, 30 μL for the first application and then 35 µL of each test product or references was applied on top of the nail with a pipette to cover the entire surface of the apical bovine nail. Care was taken to ensure that the tested product did not leak around or underneath the nail disc and that it was not applied excessively during the experimental period. Tissues were re-incubated at 30 °C with 2% CO_2_ in a saturated humidity incubator after each application from D6 to D10.

### 2.4. GMS Staining, Microscopic Data Acquisition and Semi-Quantitative Evaluation of T. rubrum Growth

Three tissue sections fixed in 10% formalin solution and embedded in paraffin blocks, each measuring 5 µm and composed of biological quadruplicates, were prepared and subsequently stained with the Grocott Methenamine Silver (GMS) method [15] using the GMS kit (Grocott—kit 01GRC100T, Histo-line Laboratories, Milan, Italy) according to the manufacturer’s protocol.

The entire histological sections were acquired at 20× magnification with the LEICA DMi8 THUNDER imager 3D microscope equipped with a camera DFC 450 C camera (brightfield, Leica Microsystem, Wetzlar, Germany), using Tilescan technology. The *T. rubrum* signal was quantified by LASX software (Version 3.0.1), which identifies fungi distributed along the tissue section with a specific threshold. The software highlights the *T. rubrum* signal by binary mask (processed image) and semi-quantifies it using the sum intensity/covered area as a parameter. A high value of sum intensity/covered area reflects a high presence of the fungus.

The average and the mean values of sum intensity/covered area of the three biological replicates (full tissue acquisition by Tilescan) were calculated considering the standard deviation.

### 2.5. Scanning Electron Microscope Analysis

The standard SEM method (Service Biotech protocol (PRO code SEM standard) of fixation was used to allow appropriate preservation of the nail structure, which had already been treated and analyzed by GMS. Thanks to this fixation and pre-treatment procedures, deformations of the nail morpho-structural characteristics potentially caused by the processing of the samples were avoided. All samples were treated with Osmium (Osmium tetroxide lot. N° BCCX9184—REF ARV01967), washed with PBS solution one time, and then treated with alcohol gradient varying in ascending order from 30° to 90°. Then, samples were passed into critical point CO_2_ at 31 °C and 73 atm (Leica EM CPD300, Leica Microsystem, Wetzlar, Germany), mounted on stubs and coated with Desk V HP and Desk V TSC sputter coater in gold-palladium. Then all mountings were visualized under the FEI Nova 450 NanoSem microscope (FEI, Hillsboro, OR, USA) under different magnitudes (1000×, 5000×, and 10,000×). Pictures were recorded for further analysis.

### 2.6. Statistical Analysis

All values are expressed as mean ± standard error of the mean. Statistical analysis of sum intensity data was performed using a one-way analysis of variance, followed by the post hoc Tukey’s test. *p*-values less than 0.05 were considered statistically significant. Statistical analyses were performed by Prism 9 (Version 10.1.1).

## 3. Results

The in vitro 3D model was successfully colonized by *T. rubrum*, and the incorporation of bovine nail sheets enhanced the microenvironment, effectively replicating the pathological condition of onychomycosis observed in vivo. Using GMS and SEM analyses, it was possible to analyze the proliferation of the fungus on the nail and tissue during the colonization phase and the effect of the infection on the nail structure.

### 3.1. GMS Staining

Microscope acquisitions of non-colonized and colonized RHE tissues were obtained for the different experimental arms (Figure 3). No staining was observed in the acquisition of the NC (Figure 3a), while a massive staining was observed in the acquisition of CNA (Figure 3b), confirming the presence of *T. rubrum* in the model. Fungi were visually less present on the colonized-RHE with one daily UDN application on the bovine nail (Figure 3c) and on the colonized-RHE with one daily EXlor application on the bovine nail (Figure 3d) as compared to CNA. In contrast, fungi were visualized on the colonized RHE with one daily PDermP application on the bovine nail (Figure 3e) and on the colonized RHE with one daily PdermP + PdermB application on the bovine nail (Figure 3f).

### 3.2. Fungus Semi-Quantitative Evaluation

Microscope acquisitions were processed using LASX software, converting the black staining into a measurable signal intensity that quantified the presence of *T. rubrum*. Sum intensities per covered area by the fungus stained across the different experimental arms after 4 days of applications are provided in Figure 4. The highest values of sum intensity/covered area were observed for the colonized RHE with no application (CNA). Compared to CNA, sum intensities were statistically significantly lower in C/UDN (−34%, *p* < 0.001) andC/EXlor (−28%, *p* < 0.020). No statistically significant difference was observed in C/PdermP (−2%) and C/PdermP + PdermB vs. CNA.

### 3.3. SEM Analysis

The appearances of the fungus hyphae, spores, and nail structure with and without one daily application of either UDN, EXlor, or Pderm products were examined by SEM. Figure 5, left panel, magnification 1000×, depicts the lamellar organization of the nail terminal part consisting of multiple layers of corneocytes cemented together and organized into lamellae on non-colonized systems (NC). A more detailed view of the structural organization of the non-colonized nail is observed at magnification 5000×. At magnification 10,000× the physiological pleated organization of this tissue is observed. Conversely, in the sample colonized with no product application (Figure 5, right panel), *T. rubrum* colonized the nail, forming a tightly meshed network of hyphae that appears fully organized in the nail depth, invading the underlying tissue. The nail keratin structure appears severely damaged. At 5000× and 10,000×, sporigenous buds on *T. rubrum* hyphae are observed, indicating a good proliferative and colonization activity.

After repeated applications of UDN (Figure 6a), the nail presents a well-preserved keratin structure without lesions compared to colonized control (CNA, visible at low magnification shown in Figure 5b). At 5000× magnification (Figure 6b), the nail is morphologically similar to the negative control (Figure 5c, NC). The product’s adherence to the nail surface is observed (Figure 6b, black arrows). At 10,000×, *T. rubrum* infection appears reduced compared to the colonized control with no application (CNA): hyphae are reduced in number and in capacity to invade the keratins layers. Furthermore, the nail structure appears preserved (Figure 6c).

With one daily application of EXlor, at 1000× magnification, the nail surface appears severely damaged: the physiological keratin structural organization is lost, and the presence of *T. rubrum* is rare and almost undetectable (Figure 7a, black arrows). At higher magnification (Figure 7b, 5000×), the presence of amorphous formations, attributable to the fungal spores (as observed in control colonized series CNA, Figure 5b), is visible (Figure 7b, black arrow). At the highest magnification (Figure 7c), the opacification of the structures under observation is visible, probably due to the residual deposit of the administered product on the spore. Nevertheless, the image magnification appears reduced compared to the control sample (black arrows).

After repeated applications of Poderm Purifying, damage to the nail’s keratin lamellar structure, likely due to a possible toxic action of the applied product, is observed at low magnification (Figure 8a, 100×). At 1000× (Figure 8b,c), *T. rubrum* is homogeneously present, forming hyphae network similar to the CNA control, suggesting that the one daily application of the product could severely damage the nail structure. Increased magnification images (Figure 8d) show holes within the lamellar body of the nail structure and the filaments of the damaged nail keratin fibers are visible at 10,000× (Figure 8e, orange arrows). With one daily application of Poderm Purifying + Poderm Booster, a very thick and dense film completely covers the nail surface, making the acquisitions not exploitable.

## 4. Discussion

In the present study, an in vitro 3D model consisting of a reconstructed human epidermis infected with *T. rubrum* and covered with a bovine nail was used to simulate in vivo onychomycosis and evaluate the effects of several commercial antifungal products with different compositions on *T. rubrum* growth. *T. rubrum* growth was evaluated through a semi-quantitative evaluation of the fungal presence using GMS analysis after one daily application of the commercial products for four days or no application (negative control). SEM analyses were performed to assess the presence of the fungi beneath the nail and the effects of repeated product applications on the nail structure.

The negative controls demonstrate that RHE infection and fungal growth evaluated using the 3D model are evidenced by a high value of sum intensity/covered area of GMS staining. Additionally, the presence of the hyphal network and damaged nail structure, as observed by SEM, further supports these findings. Alterations of the internal nail microstructure have been observed in onychomycotic nails compared to healthy nails [16]. A significant reduction of disulphide bonds was observed in onychomycotic nails leading to keratin disorganization and increased nail porosity. The SEM observations from our in vitro system align with the in vivo alterations of the nail in onychomycosis. This demonstrates that our in vitro 3D model is effective for simulating onychomycosis and for evaluating the antifungal effects.

Four topical products with various compositions were applied once daily onto the bovine nail. Different effects on *T. rubrum* growth and on the structure of the bovine nail were observed. A statistically significant lower *T. rubrum* growth was observed after the applications of Urgo Damaged Nails or Excilor Forté compared to the control group, which consisted of the colonized RHE that did not receive any treatment. No significant difference was observed between the applications of the two products. Additionally, no effect of Poderm Purifying on fungal growth was found, and no synergistic action was found between Poderm Purifying and Poderm Booster. The antifungal action of essential oils (EO) has been previously reported, suggesting that EO could be promising alternative candidates for the management of onychomycosis [17,18,19,20]. In the present study, we did not observe any effect of Poderm Purifying applications, despite this product mainly containing natural seeds oils, floral oils, and monoterpenes such as geraniol, citronellol, and linalool found in essential oils. A strong antifungal activity of geraniol and citronellol have been shown in vitro with a susceptibility test using the broth microdilution technique [21]. However, such in vitro test systems are less complex as the organized keratin network is not present, compared to the in vitro human nail model or the 3D model used in the present study. In microdilution or even with agar dilution techniques [22], the interaction of the active molecules with the microorganisms is facilitated, allowing for the antifungal effect to be observed more readily. Therefore, it is likely that repeated applications of such active molecules in a more complex model, such as one comprising a colonized RHE covered with a keratin barrier, may not have any inhibitory effect on the fungus growth.

In addition to the model’s ability to reflect effects of antifungal agents on fungal growth and invasion, it also captures potential side effects of topical treatments. For instance, several signs of compromised nail structure were evidenced after repeated applications of the product containing natural oils and monoterpenes. SEM analyses revealed alterations in the lamellar structure of nail keratin, degradation of nail keratin fibers, and even the presence of holes after four days of daily application. Essential oils have been shown to induce a damaging effect on the skin stratum corneum structure due to conformational changes of the lipid and keratin network in the stratum corneum [23], resulting in a change of the orderly and compact structure that increases the skin permeability and reduces the effect of barrier function. While the lipid and keratin content are different between skin stratum corneum and nail structures [24], our observation of the delamination of the keratin network in the bovine nail aligns with the previous report indicating a damaging effect on skin stratum corneum structure and also with the use of monoterpenes, especially linalool, as highly effective chemical penetration enhancers for the transungual delivery of antifungal agents [25]. Thus, considering the need for long-term treatment duration to achieve mycological cure, repeated use of monoterpene-based products may seriously harm the nail structure and hinder normal nail regrowth. In addition, as terpenes such as limonene, linalool, and geraniol are known flagrance allergens [26], an increased risk of developing allergies may be observed in case of repeated accidental applications of such products on the skin surrounding the nail. Therefore, products that are safe to use and do not impede nail regrowth should be prioritized when selecting appropriate treatment options for individuals with onychomycosis. In contrast, repeated applications of Urgo Damaged Nails help preserve the lamellar keratin structure of the nail, similar to that of a healthy, non-colonized nail. No structural damage is observed, suggesting that this product may be suitable for long-term repeated applications and effective in protecting the nail during the regrowth process.

All products but the Poderm Booster were applied according to the recommended posology for in vivo use. Recommendations of Poderm Booster applications are twice a week, concomitantly with one daily application of Poderm Purifying (rebranded as Sérum Mycose des ongles Poderm^®^). In the present study, it was applied once daily for four days on top of Poderm Purifying. All the colonized RHE covered with a bovine nail received daily the same volume of product using the same experimental conditions, allowing for the comparison of observed effects. The applied volumes onto the bovine nail daily were adequate as no overflow, even with the concomitant applications of the two Poderm products, was observed.

Model limitations such as its unsuitability for long-term studies, lack of validation with additional fungal species, and potential differences between bovine and human nails should be considered when interpretating these results. Due to the high sensitivity of in vitro 3D model with human tissue, an increased number of applications or longer treatment duration were not considered to ensure the viability of the system. The 3D model was colonized with *T. rubrum*, the most representative trichophyton strain of onychomycosis. As different colonization capacities of Trychophyton strains were evidenced in vitro on bovine membranes [27], it would be interesting to evaluate the ability of other trichophyton such as *Trichophyton mentagrophytes* to grow, invade the 3D model, and respond to antifungal agent.

## 5. Conclusions

The in vitro 3D model used in the present study serves as an effective preclinical model for ensuring fungal attachment and accurately simulates the pathological conditions of the nail in the presence of onychomycosis, mirroring the conditions observed in humans. This model is composed of *T. rubrum*-infected RHE covered with bovine nail sheet and supports for nail fungal infection with the presence of a moist and metabolic active surface to create relevant biological niche for fungal growth. The effects on fungal growth and nail structure were demonstrated by testing different products, suggesting that this in vitro 3D model of onychomycosis could be used to predict the effects of antimycotic treatment in humans. However, long-term clinical studies are needed to evaluate the effect of these products on subjects with moderate onychomycosis.

## Figures and Tables

**Figure 1 jof-11-00285-f001:**
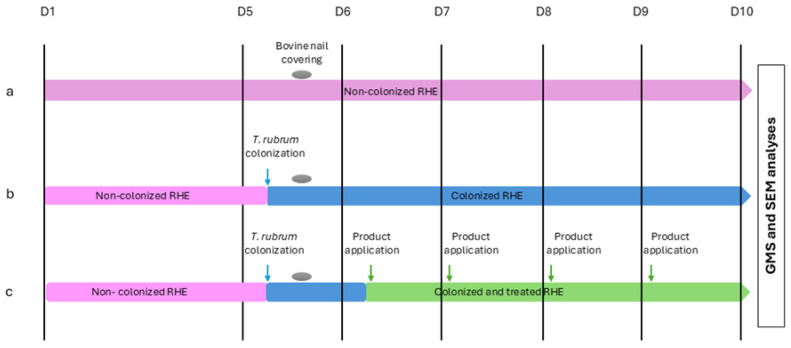
Set-up of the 3D model. (**a**): non-colonized RHE; (**b**): colonized RHE with no product application; (**c**): colonized RHE with product applications on the bovine nail. blue arrow: colonization with *T. rubrum*, grey oval: Bovine nail sheets covering of the RHE, green arrow: once daily application of the product.

**Figure 2 jof-11-00285-f002:**
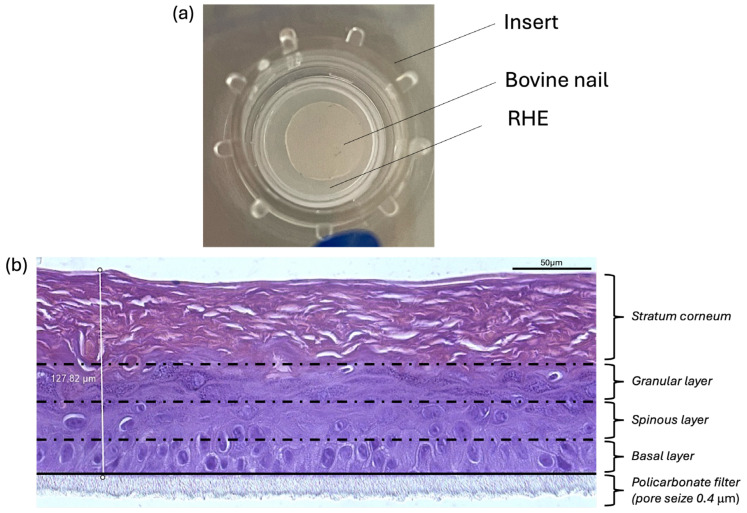
(**a**) Representation of the 3D model of SkinEthic^TM^ Reconstructed Human Epidermis (RHE) covered by a bovine nail sheet, (**b**) Histological observation of the non-colonized RHE.

**Figure 3 jof-11-00285-f003:**
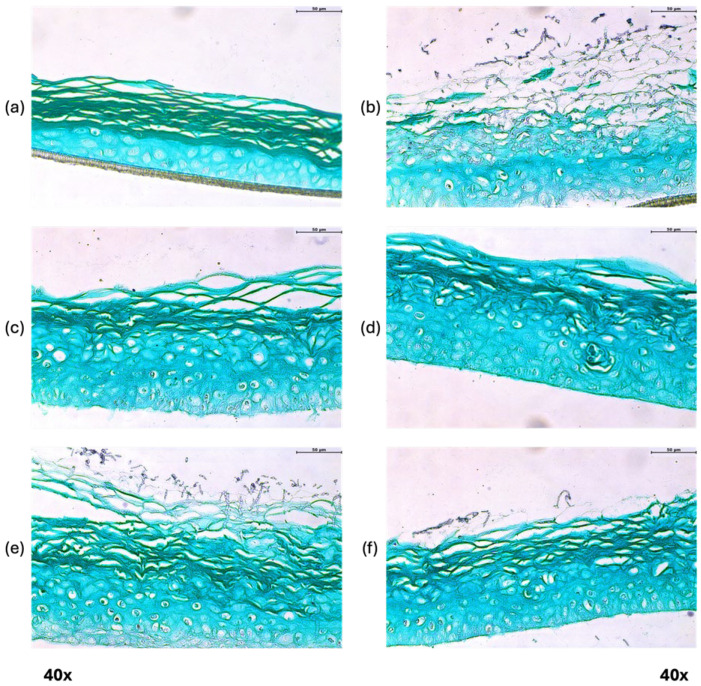
Microscope acquisitions (scale bar = 50 µm) of GMS stained RHE. (**a**) Non-colonized RHE (NC); (**b**) Colonized RHE with no product application (CNA); (**c**) colonized RHE after UDN applications (C/UDN); (**d**) colonized RHE after Excilor applications (C/EXlor); (**e**) colonized RHE after Poderm Purifying applications (C/PdermP); (**f**) colonized RHE after Poderm Purifying + Poderm Booster applications (C/PdermP + PdermB). Fungus cell walls and debris are stained in brown to black. RHE is stained in green.

**Figure 4 jof-11-00285-f004:**
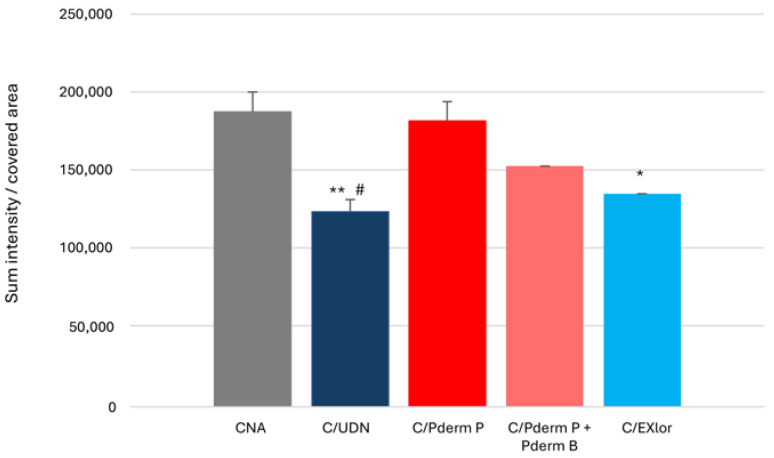
Semi-quantitative evaluation of *T. rubrum* in colonized RHE. *T. rubrum* semi-quantification mean results are expressed as Sum intensity/covered area by the fungus stained with GMS. Results are expressed as mean ± standard error of the mean of triplicates. * *p* < 0.02 vas. CNA, ** *p* < 0.001 vs. CNA, # *p* < 0.02 vs. C/Pderm P.

**Figure 5 jof-11-00285-f005:**
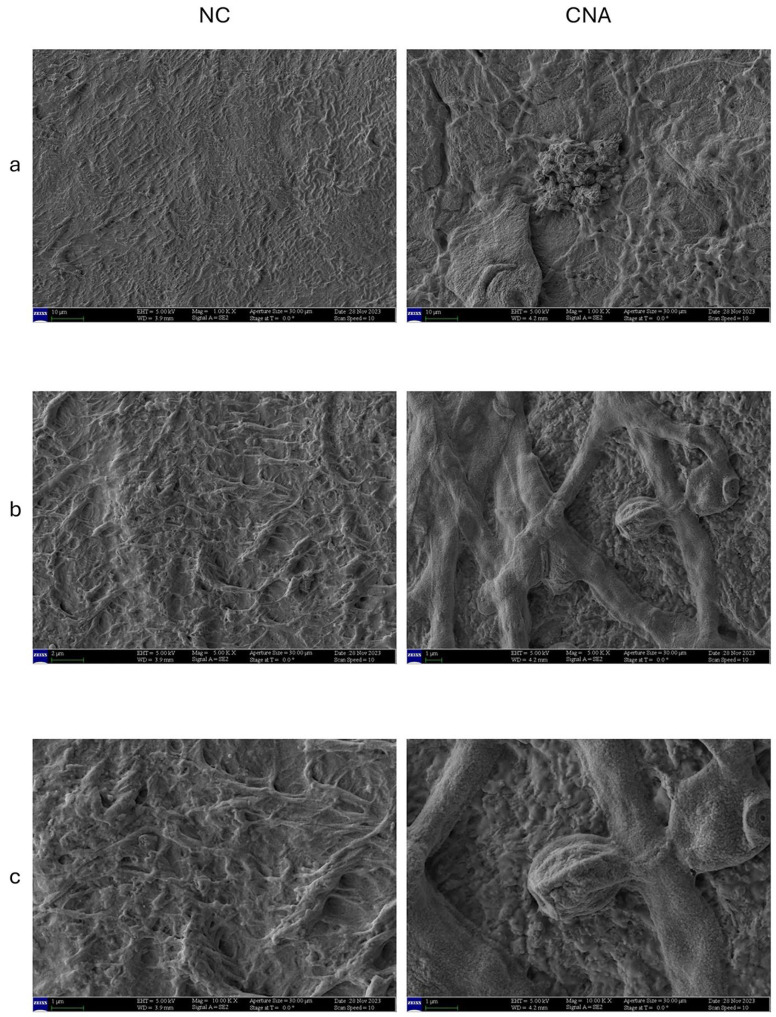
SEM acquisition of GMS stained RHE covered with the bovine nail. Left panel: Non-colonized RHE (NC); Right panel: Colonized RHE with no product application (CNA). Acquisitions were obtained at several magnifications: (**a**) at 1000×, (**b**) at 5000×, and (**c**) at 10,000×. In (**b**,**c**) CNA panels, an amorphous agglomerate is visible, which may be attributable to the presence of fungal spores.

**Figure 6 jof-11-00285-f006:**
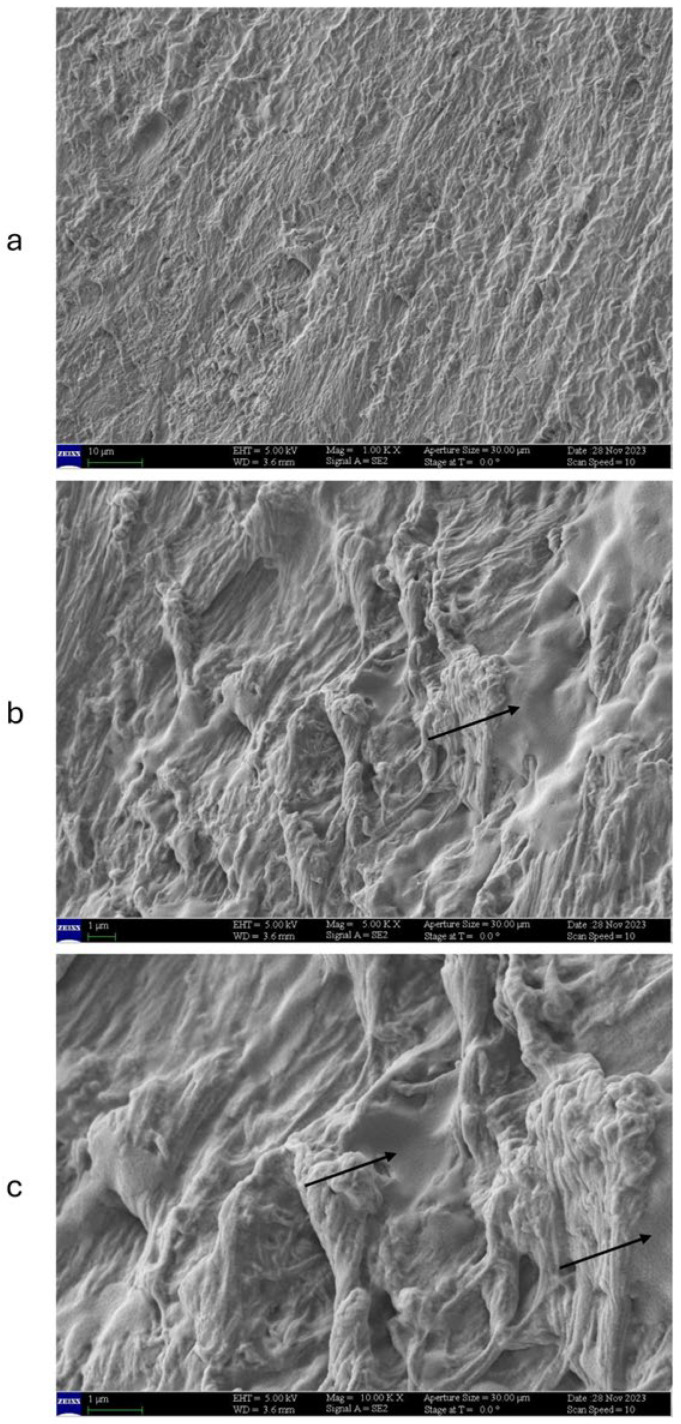
SEM acquisition showing the growth of the hyphal network that is covered with the bovine nail receiving one daily application of UDN. Acquisitions were obtained at several magnifications: (**a**) at 1000×, (**b**) at 5000×, and (**c**) at 10,000×. Traces of applied products on RHE surface are visible (black arrows).

**Figure 7 jof-11-00285-f007:**
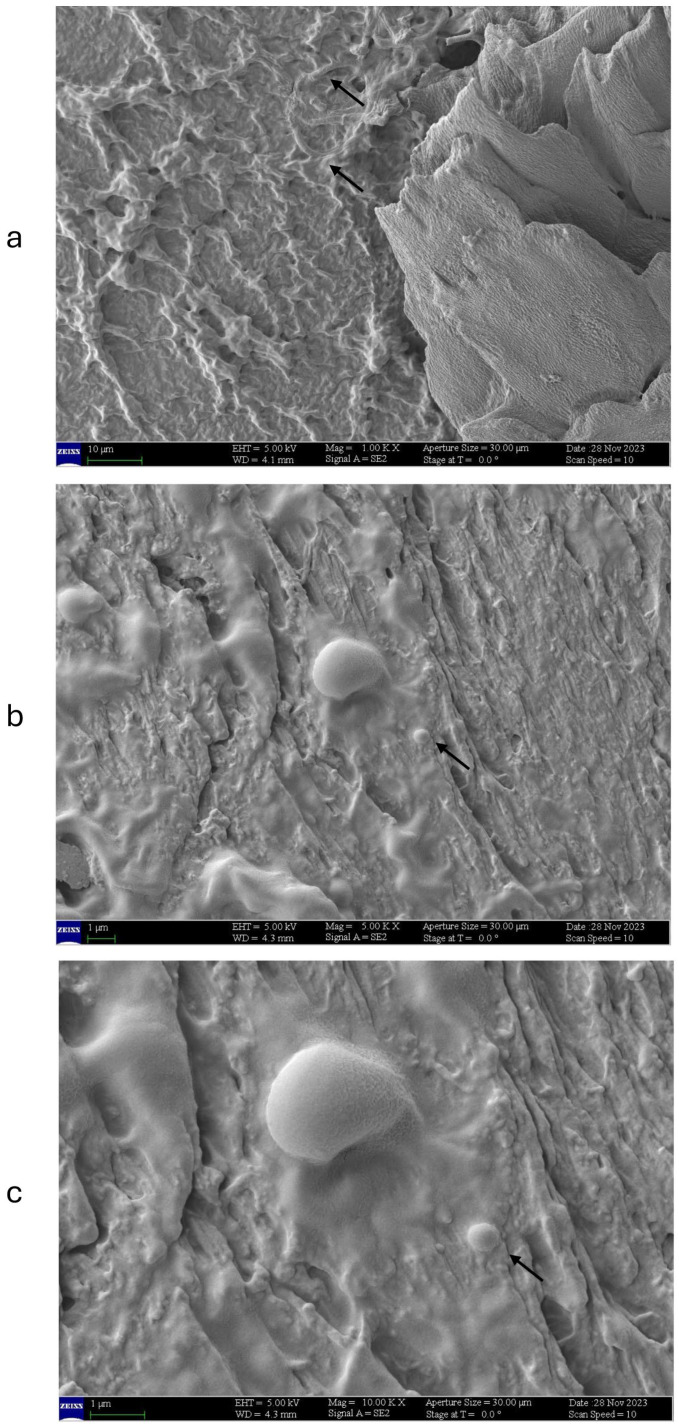
SEM acquisition showing the growth of the hyphal network that is covered with the bovine nail receiving 1×/day applications of Excilor Forté. Acquisitions were obtained at several magnifications: (**a**) at 1000×, (**b**) at 5000×, and (**c**) at 10,000×. Black arrow: presence of fungal spores.

**Figure 8 jof-11-00285-f008:**
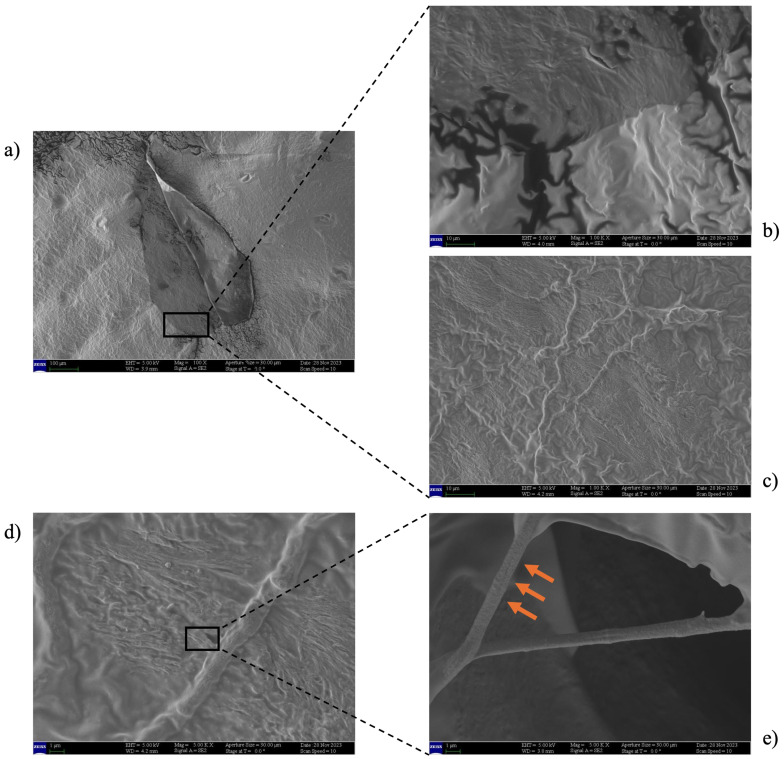
SEM acquisition showing the growth of hyphal network that is covered with the bovine nail receiving 1×/day application of Poderm Purifying. Acquisitions were obtained at several magnifications: (**a**) at 100×, (**b**,**c**) at 1000× (**d**) at 5000×, and (**e**) at 10,000×. Orange arrows show damaged nail keratin fibers.

## Data Availability

Due to privacy, raw research data are not available. Information may be provided to interested researchers upon request to the corresponding author.

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
