# Peer review of "Use of a 3D Model with Reconstructed Human Epidermis Infected with Fungi and Covered with a Bovine Nail to Simulate Onychomycosis and to Evaluate the Effects of Antifungal Agents"

_jof, 2025, doi:10.3390/jof11040285_

Round 1

Reviewer 1 Report

other antifungals compounds can be used in the experiments

please, cited the figure 2b in the text 

Author Response

  1. Please clarify information about the antifunglas and bovine nail used. Are the products used specific to your country? Why didn’t you use an antifungal like fluconazole?

The antifungals used in the present study are commercially available products in Europe. They are all classified as medical devices and are CE-marked. We specifically chose to test medical devices rather than conventional antifungal drugs. Our primary goal was to investigate the mechanical antifungal activity of the medical device while avoiding products with pharmacological activity such as medicinal drugs containing fluconazole or terbinafine. Furthermore, French regulations do not permit direct comparisons of effects between medical devices and medicinal drugs, we deliberately did not include any medicinal drugs in our 3D model testing.

The decision to apply bovine nail sheets to the reconstructed epidermis model (SkinEthic™ RHE) was made to better mimic human nails, which are naturally attached to fingers.

  1. Please clarify information about the methodology of antifunglas used

We assume that the reviewer is requesting more details about the tested products. However, as explained in the cover letter, the nature of the information we can provide is limited due to regulatory constraints. Nonetheless, we have revised Section 2.1, page 2/15 of the Methods section by adding the following sentences:
“All tested products are commercial formulations designed to strengthen and restore nail structure damaged by onychomycosis. All products are classified as medical devices.”

  1. Please explain the figure 2 b in the result.

Figure 2b represents the histological observation of a healthy reconstructed human epidermis. As suggested, we have improved Figure 2b by adding annotations to indicate the different epidermal layers. The revised Figure 2 has been resubmitted.

Additionally, we have added the following text to the Methods section:
“Figure 2b shows a histological observation of the healthy RHE, with the stratum corneum and the different epidermal layers clearly visible.”

  1. other antifungals compounds can be used in the experiments

We acknowledge that many other antifungal compounds could have been tested in this experiment. Commercial topical antifungals are available as cosmetic products, medical devices, or medicinal drugs. In our study, we specifically tested CE-marked medical devices designed to strengthen and restore nail structure damaged by onychomycosis.

We selected Excilor as the positive control to validate the experiment, based on its proven antifungal efficacy in both in vitro and in vivo studies. The rationale for selecting Excilor as the positive control has now been added to Section 2.1:
“Excilor was chosen as the positive control because its daily application has been shown to have a similar effect to once-weekly amorolfine application in vitro [13]. Additionally, its clinical efficacy is being evaluated in subjects with onychomycosis [14].”

As mentioned in response to question #1, we are also constrained by regulatory limitations regarding product comparisons.

New references (13 and 14) have been added:
13. Sleven R, Lanckacker E, Delputte P, Maes L, Cos P. Evaluation of topical antifungal products in an in vitro onychomycosis model. Mycoses 2016, 59, 327-330.
14. Clinical evaluation of efficacy and safety of a medical device for the treatment of toenail onychomycosis. https://www.excilor.com/fr-fr/product/excilor-traitement-de-la-mycose-de-l-ongle-forte. Accessed on 28/02/2025.

  1. please, cited the figure 2b in the text.

See response to question #3.

Reviewer 2 Report

It is an interesting contribution in terms of innovation, as it offers a reliable practical model, suitable for better understanding the etiopathogenesis and treatment of onychomycosis.

It is an interesting contribution in terms of innovation, as it offers a reliable practical model, suitable for better understanding the etiopathogenesis and treatment of onychomycosis.

The manuscript is well written and presents a robust set of results. Only a few issues

  • The authors need to provide more information about the 3D model. If it is new and was created for the present study, the authors should present, in the methods, how the model was created. On the other hand, if it is a model already available and used in other studies, these should be cited as a reference, the same way if there is a patent number. Or is it a commercial product, which has been purchased? Please clarify.
  • I did not find information about the 4 products tested (Urgo Damaged Nails, Excilor, Poderm Purifying, Poderm Booster), are they antifungal? It would be good for the reader to understand this issue, authors can include it either in methods or in the discussion.
  • In methods, separate each subsection with a line
  • The caption in fig 5 is confusing, it would be better to insert the respective magnification bars in each photo and thus eliminate them from the text.
  • The caption in fig 6: It's very strange "SEM acquisition of GMS stained" I suggest removing "GMS stained" and replacing it with something like hyphae, fungal growth or another term to that effect

Page 3: I assume there is a typo “2.23. D model....”

Page 3: The authors claim that “30 μL of T. rubrum spores were used to colonize the entire tissue surface”, but they did inform about the inoculum concentration, it is essential for future quantitative evaluation.

Page 6: 3.2. Fungus quantification: In my opinion this term is not appropriate, since there is no real quantification but rather an estimate through the area covered (this was just inferred), so I suggest replacing. Likewise, please consider the caption in fig 4 (page 7).

Author Response

It is an interesting contribution in terms of innovation, as it offers a reliable practical model, suitable for better understanding the etiopathogenesis and treatment of onychomycosis.

The manuscript is well written and presents a robust set of results. Only a few issues

  • The authors need to provide more information about the 3D model. If it is new and was created for the present study, the authors should present, in the methods, how the model was created. On the other hand, if it is a model already available and used in other studies, these should be cited as a reference, the same way if there is a patent number. Or is it a commercial product, which has been purchased? Please clarify.

This model is new and was specifically developed for this study. The SkinEthic™ RHE and bovine nail sheet are commercially available tissue systems, distributed by Episkin (Lyon, France) and XENOMETRIX AG (Allschwil, Switzerland), respectively, as indicated in the Materials and Methods section. The innovation lies in combining these two tissues to replicate the physiological onycho-structure and recreate the natural microenvironment. Details on this combination are provided in Section 2.2 and are briefly illustrated in Figure 1, which has been revised accordingly.

Although innovative, like all tissue models developed by Vitroscreen, this 3D model simulating human onychomycosis is not patented. The revised Figure 1 has been resubmitted with the revised manuscript.

  • I did not find information about the 4 products tested (Urgo Damaged Nails, Excilor, Poderm Purifying, Poderm Booster), are they antifungal? It would be good for the reader to understand this issue, authors can include it either in methods or in the discussion.

Please refer to the response to question #1 from Reviewer 1.

  • In methods, separate each subsection with a line

Section “2.2. Set-up of the 3D model of colonized Reconstructed Human Epidermis covered by a bovine nail” has been revised and simplified with all subsection titles removed.

  • The caption in fig 5 is confusing, it would be better to insert the respective magnification bars in each photo and thus eliminate them from the text.

All SEM images already include explanatory magnification bars. However, due to the reduced size of images in Figure 5, the bars may not have been clearly visible. The figure has been reworked to present larger SEM images, improving visibility. Similarly, Figures 6 and 7 have also been revised.

We hope these adjustments meet the reviewer’s expectations. If not, we believe it is best to retain magnification details in both the figure legend and text.

Note: All original high-resolution SEM images were submitted with the initial manuscript. The revised Figures 5, 6, and 7 have been resubmitted with the revised manuscript.

  • The caption in fig 6: It's very strange "SEM acquisition of GMS stained" I suggest removing "GMS stained" and replacing it with something like hyphae, fungal growth or another term to that effect

The caption of Figure 6 has been revised as suggested.  It now reads: “Figure 6. SEM image acquisition showing the growth of the hyphal network that is covered with the bovine nail receiving 1x/day application of UDN.

Consequently, the captions of Figure 7 and Figure 8 have also been revised for consistency:

Figure 7. SEM image acquisition showing the growth of the hyphal network that is covered with the bovine nail receiving 1x/day application of Excilor Forté.

Figure 8. SEM image acquisition showing the growth of hyphal network that is covered with the bovine nail receiving 1x/day application of Poderm Purifying.

Page 3: I assume there is a typo “2.23. D model....”

The typo is corrected. It now reads: “2.2. Set-up of the 3D model of colonized Reconstructed Human Epidermis covered by a bovine nail”.

Page 3: The authors claim that “30 μL of T. rubrum spores were used to colonize the entire tissue surface”, but they did inform about the inoculum concentration, it is essential for future quantitative evaluation.

The initial OD measurement could give an idea of the inoculum concentration. For completeness, we have also included the data related to viable count (1.1 x 10⁷ CFU/mL, at page 3).

Page 6: 3.2. Fungus quantification: In my opinion this term is not appropriate, since there is no real quantification but rather an estimate through the area covered (this was just inferred), so I suggest replacing. Likewise, please consider the caption in fig 4 (page 7).

The reviewer is correct, as no direct quantification of fungal growth was conducted. Instead, a semi-quantitative evaluation was performed. The caption of Figure 4 (page 7) and Section 3.2 (page 6) have been revised accordingly.

The term "quantification" has now been replaced with "semi-quantitative evaluation" throughout the text.

Reviewer 3 Report

Section 2.5: Because the title claims that the study evaluated “the effects of antifungal agents”, the authors should have included an approved antifungal agent (e.g., terbinafine hydrochloride nail lacquer) for comparison and thorough model validation. All products described in the manuscript seem to be cosmetic products to strengthen/restore the nail structure but are not actual antifungal agents. It is possible that the mechanical impact of product application and/or the products’ biochemical properties or their propensity to seal the nail surface impaired fungal growth without actually eliciting antifungal activity. For proper model validation, and to support the claim in the title or the claim of “mycological cure” in the Discussion, testing of an actual antifungal agents would be needed. Including a true antifungal agent would also help to alleviate concerns about industry affiliations of two authors with the company producing the UDN agent that yielded the most favorable data in the present study.

In addition, there is a quite noticeable discrepancy between the relative amount of fungal material seen on Figure 3 and the quantitative results in Figure 4. For instance, image 3c shows no fungal material whereas Figure 4 suggests only about a 30% reduction for that group, with little variation. Therefore, the authors should check their image analysis algorithms for background noise and/or select more representative images for Figure 3. Given that model reproducibility is a key criterion for a new model, the authors should display individual replicates as data points within Figure 4.

Most of the second paragraph of the Introduction (everything after REF 10) is rather unrelated to the content of the manuscript which focuses on Trichophyton rubrum. The authors should consider tailoring the Introduction more closely to the content of the manuscript. In particular, the authors might want to expand their introduction of existing models of onychomycosis as well as their advantages and disadvantages to better define the need for their new model.

Figure 1 seems overly complicated. For instance, what do the two shades of blue mean? The schematic should be simplified. All introductions used in the figure should be introduced in the figure legend.

The numbering (and partial lack thereof) of sub-sections in section 2.23 (?) should be revised.

Section 2.3: How was the test product applied? Just pipetted on top of the nail or applied using a small brush?

Section 2.4: Given that GMS staining is a common method in mycology, its extensive description in the Methods section is not needed. It would suffice to specify the kit and vendor, along with a statement whether the authors followed the manufacturer’s protocol or amended the method for their study. The abbreviation GMS needs to be introduced.

Sub-header 2.5 should be changed to incorporate the actual method (e.g., Microscopic data acquisition and quantification of T. rubrum). In addition, the authors need to introduce the active ingredients of these products.

Results section: Before discussing downstream readouts, the authors should comment at the beginning of the Results section how efficient their model setup was, e.g., state whether technical challenges were encountered.

Section 3.1: The extensive description of color is not needed. The authors can just refer to the amount of fungal material and mention once (in the text or figure legend) that it is indicated by brown/black color.

Figure 3: Several images do not contain a scale bar.

The size of the images in Figures 6 and 7 should be increased by utilizing the full width of the page.

What is meant by “dimensions” when describing Figure 7c? The extent of hyphal material?

Figure 8: The reference in the text needs to be changed from 7b to 8b.

First paragraph of the Discussion: How did the authors evaluate fungal growth “beneath the nail” using SEM? It seems that the images rather capture the surface of the nail.

The forth paragraph of the Discussion should be written more concisely. It should also be preceded by a better transition, e.g., “In addition to the model’s ability to reflect effects of therapeutic agents on fungal growth and invasion, it also captures potential side effects of topical treatments. For instance, several signs of compromised nail structure were evident after…”.

The Discussion should include a summary of limitations, e.g., with regards to the model’s unsuitability for long-term studies, lack of validation with additional fungal species, potential differences between bovine and human nails etc.

The manuscript is well written but still requires thorough editing by a native English speaker due to several syntax and grammar issues (e.g., use of articles) and some unusual/non-scientific phraseology (e.g., “thanks to” in section 2.6).

Author Response

Reviewer 3

 Section 2.5: Because the title claims that the study evaluated “the effects of antifungal agents”, the authors should have included an approved antifungal agent (e.g., terbinafine hydrochloride nail lacquer) for comparison and thorough model validation. All products described in the manuscript seem to be cosmetic products to strengthen/restore the nail structure but are not actual antifungal agents. It is possible that the mechanical impact of product application and/or the products’ biochemical properties or their propensity to seal the nail surface impaired fungal growth without actually eliciting antifungal activity. For proper model validation, and to support the claim in the title or the claim of “mycological cure” in the Discussion, testing of an actual antifungal agents would be needed. Including a true antifungal agent would also help to alleviate concerns about industry affiliations of two authors with the company producing the UDN agent that yielded the most favorable data in the present study.

This comment aligns with questions 1 & 2 of Reviewer 1. The antifungals used in the present study are commercially available medical devices and are CE-marked in Europe.

As rightly pointed out by the reviewer, all these products are designed to strengthen and restore the nail structure, with two of them specifically claiming to treat nail mycosis.

We chose to test similar medical devices rather than conventional antifungal references. Our primary objective was to investigate the mechanical antifungal activity of these medical devices while avoiding products with pharmacological activity, such as medicinal drugs containing fluconazole or terbinafine. Furthermore, French regulations do not allow direct comparisons between the effects of medical devices and medicinal drugs. Therefore, we deliberately excluded medicinal drugs such as terbinafine nail lacquer or fluconazole when testing and validating the 3D model.

Instead, we selected the Excilor product as the positive control based on its proven antifungal effect in vitro and in vivo, considering it a true antifungal product (see response to question 4 from Reviewer 1).

As indicated in Section 6, our objective is to propose a 3D model that simulates human onychomycosis and could serve as a predictive tool for assessing the effects of antimycotic treatments in humans. However, mycological cure claims for topical antifungal products can only be substantiated through long-term clinical evaluations, not solely based on results obtained using the in vitro 3D model.

In addition, there is a quite noticeable discrepancy between the relative amount of fungal material seen on Figure 3 and the quantitative results in Figure 4. For instance, image 3c shows no fungal material whereas Figure 4 suggests only about a 30% reduction for that group, with little variation. Therefore, the authors should check their image analysis algorithms for background noise and/or select more representative images for Figure 3. Given that model reproducibility is a key criterion for a new model, the authors should display individual replicates as data points within Figure 4.

The images shown in Figure 3 are histological acquisitions of the most representative sections of the paraffin-embedded tissue.

As a result, signal quantification across different sections may exhibit natural variability due to the assessment of different transverse sections of the same sample. These images provide a visual evaluation of the effects of the tested product.

The results presented in Figure 4 represent the mean signal sum intensity per covered area, providing a semi-quantitative assessment of the signal based on the entire tissue section acquired using Tilescan technology. This methodological difference explains the discrepancy between the two results.

To ensure greater consistency with the signal quantification histograms reported in Figure 4, we have selected a different image for the CNA series in Figure 3b.

Most of the second paragraph of the Introduction (everything after REF 10) is rather unrelated to the content of the manuscript which focuses on Trichophyton rubrum. The authors should consider tailoring the Introduction more closely to the content of the manuscript. In particular, the authors might want to expand their introduction of existing models of onychomycosis as well as their advantages and disadvantages to better define the need for their new model.

The introduction is rewritten following the reviewer’s suggestion.  The second paragraph now focuses on existing in vitro and ex vivo models designed to evaluate antifungal agent permeation, and for some models the inhibitory effect on fungus growth. The revised manuscript now reads on page 2/16:

Many in vitro models and ex-vivo models have been developed in the last decade to assess the penetration and effects of antifungal drugs topically applied to either human nails mounted in specific chambers seeded with Trichophyton rubrum, bovine hoof plates, or artificial nail plates such as keratin bio-membranes generated from human hair [9-12]. While most of them mainly evaluate the permeation capacity of antifungal agents, their inhibitory effects on fungus growth were assessed with models having fungi grown on the surface of the artificial nails and not beneath the nail which is less of the human onychomycotic nail. Moreover, most in vitro models do not repro-duce the microenvironment of the nail bed which provides viable conditions for fungal growth.

Thus, we aimed to develop a 3D model that provides appropriate conditions for fungal growth and simulates a human onychomycosis. To best recreate an in vitro environment suitable for fungal infection, we selected a model that included (a) a 3D reconstructed human epidermis (RHE) in vitro, which represents an interesting test system, due to its morphological similarity and metabolic activity to the skin, as well as being a fully viable substrate along with the nail ; and (b) bovine nail sheets representing an essential substrate and microenvironment for the fungal growth and adhesion. This study evaluates the suitability of such a 3D model as a preclinical onychomycosis model and assesses the effects of different products on fungal growth and nail structure.”

References 9 to 12 are now supporting the revised paragraph:

  1. Brown M, Turner R, Wevrett SR. Use of in vitro performance models in the assessment of drug delivery across the human nail for nail disorders. Expert Opin Drug Deliv 2018, 15, 983-989.
  2. Naumann S, Meyer JP, Kiesow A, Mrestani Y, Wohlrab J, Neubert RH. Controlled nail delivery of a novel lipophilic antifungal agent using various modern drug carrier systems as well as in vitro and ex vivo model systems. J Control Release 2014, 180, 60-70.
  3. Valkov A, Zinigrad M, Sobolev A, Nisnevitch M. Keratin Biomembranes as a Model for Studying Onychomycosis. Int J Mol Sci 2020, 21, 3512.
  4. Christensen L, Turner R, Weaver S, Caserta F, Long L, Ghannoum M, Brown M. Evaluation of the Ability of a Novel Miconazole Formulation To Penetrate Nail by Using Three In Vitro Nail Models. Antimicrob Agents Chemother 2017, 61, e02554-16.

Figure 1 seems overly complicated. For instance, what do the two shades of blue mean? The schematic should be simplified. All introductions used in the figure should be introduced in the figure legend.

Figure 1 has been simplified to display the key steps of the 3D model set-up and analyses. A brief description is now included in the figure legend. To better reflect the figure its content, the title has been changed to: “Figure 1. Set-up of the 3D model”.

The revised figure now appears as follows:

The following text has been added to the figure legend:

Figure 1. Set-up of the 3D model. Briefly, the reconstructed human epidermis (RHE, vertical bars) were cultured in standard growth and maintenance medium from D1 to D10 in a), b) and c). At the end of D5, in b) and c), RHEs were colonized with T. rubrum (blue arrow). Bovine nail sheets (grey oval) were applied onto all RHEs. In c, tested products (green arrow) were applied once daily on the bovine nail from D6 to D10. After D10, RHE and bovine nails were collected, stained with the GMS method and further fixed for SEM analyses of the nail appearance and structure (yellow triangle). A: non-colonized RHE ; B: colonized RHE with no product application; C: colonized RHE with product applications on the bovine nail.

Revised Figure 1 is part of the resubmission package.

The numbering (and partial lack thereof) of sub-sections in section 2.23 (?) should be revised.

The title of the section has been revised and now reads: “2.2. Set-up of the 3D model of colonized Reconstructed Human Epidermis covered by a bovine nail”. To simplify this section, subsection titles have been deleted.

Section 2.3: How was the test product applied? Just pipetted on top of the nail or applied using a small brush?

The product was applied topically with a pipette due to the bovine nails size (smaller than RHE tissue) and to allow the same volume between replicates. The details were added in the section 2.3 of Material and Methods paragraph (page 5/16). It now reads:

Except for the two controls, non-colonized RHE and colonized RHE with no application, 30μL for the first application and then 35 µL of each test product or reference was applied on top of the nail with a pipette to cover the entire surface of the apical bovine nail

Section 2.4: Given that GMS staining is a common method in mycology, its extensive description in the Methods section is not needed. It would suffice to specify the kit and vendor, along with a statement whether the authors followed the manufacturer’s protocol or amended the method for their study. The abbreviation GMS needs to be introduced.

Sections 2.4 and 2.5 of the previous manuscript have been gathered in one section: “2.4. GMS staining, Microscopic Data acquisition and semi- quantitative evaluation of T. rubrum”.

This section no longer extensively describes the GMS staining method and only references the GMS kit (Grocott – kit 01GRC100T, Histo-line Laboratories, Italia) and its use according to the manufacturer’s protocol, as suggested by the reviewer. The description of the microscopic data acquisition and image analysis used for indirect quantification of fungal presence remains unchanged.

Then all subsequent sections are renumbered.

Sub-header 2.5 should be changed to incorporate the actual method (e.g., Microscopic data acquisition and quantification of T. rubrum).

See previous answer.

In addition, the authors need to introduce the active ingredients of these products.

Since all tested products are medical devices designed to perform mechanical actions, they do not contain any active ingredients. This also clarifies why no detailed composition of the tested products is provided in section 2.1.

Results section: Before discussing downstream readouts, the authors should comment at the beginning of the Results section how efficient their model setup was, e.g., state whether technical challenges were encountered.

No significant technical challenges were encountered during the setup of the 3D model. The time required for successful and optimal RHE infection with the fungus, as well as the acceptable duration of the product application phase that ensures the viability of the RHE, were tested to define appropriate experimental conditions. These conditions are outlined in section 2.2 of the manuscript.

The following paragraph is added to the section 3 of the revised manuscript (page 6/16):

The in vitro 3D model was successfully colonized by T. rubrum, and the incorporation of bovine nail sheets enhanced the microenvironment, effectively replicating the pathological condition of onychomycosis observed in vivo. Using GMS and SEM anal-ysis, it was possible to analyze the proliferation of the fungus beneath the nail and on the tissue during the colonization phase and the effect of the infection on the nail structure.

Section 3.1: The extensive description of color is not needed. The authors can just refer to the amount of fungal material and mention once (in the text or figure legend) that it is indicated by brown/black color.

Section 3.1 is simplified in the revised manuscript (). It now reads:

Microscope acquisitions of non-colonized and colonized RHE tissues were obtained for the different experimental arms (Figure 3). No staining was observed in the acquisition of the NC (Figure 3A), while a massive staining was observed in the acquisition of CNA (Figure 3B), confirming the presence of T. rubrum in the model. Fungi were visually less present on the colonized-RHE with one daily UDN application on the bovine nail (Figure 3C) and on the colonized-RHE with one daily EXlor application on the bovine nail (Figure 3D) as compared to CNA. On the opposite, fungi were visualized on the colonized RHE with one daily PDermP application on the bovine nail (Figure 3E) and on the colonized RHE with one daily PdermP + PdermB application on the bovine nail (Figure 3F).

Figure 3: Several images do not contain a scale bar.

 Figures 3a, 3c and 3f have been revised and now include a scale bar, similar to figures 3b, 3d and 3e.

Revised figure 3 is part of the resubmission package.

The size of the images in Figures 6 and 7 should be increased by utilizing the full width of the page.

The size of all SEM images in Figure 6 and in Figure 7 has been increased according to the reviewer’s suggestions.

The size of all SEM images in Figure 5 has been increased as well.

What is meant by “dimensions” when describing Figure 7c? The extent of hyphal material?

In Figure 7c, as in the other references to Figure 7, the dimensions refer to the magnification used in the SEM images. The magnification is necessary to allow the observer to understand which are the references in terms of microns that are being used, and that even if they are present in the toolbar of the photo, it is preferable to report them in the descriptive text to underline their descriptive value.

The sentence is now rewritten as follows (page 11/16)

Nevertheless, the image magnifications appear reduced compared to the control sample (black arrows).”

Figure 8: The reference in the text needs to be changed from 7b to 8b.

Thank you for pointing this error. The text is fully revised and now reads:

At 1000X (Figure 8b, Figure 8c), T. rubrum is homogeneously present forming hyphae network similar to the CNA control, suggesting that the one daily application of the product could severely damage the nail structure. Increased magnification images (Figure 8d) show holes within the lamellar body of the nail structure and the filaments of the damaged nail keratin fibers are visible at 10,000X (Figure 8e, orange arrows).

First paragraph of the Discussion: How did the authors evaluate fungal growth “beneath the nail” using SEM? It seems that the images rather capture the surface of the nail.

The SEM analysis captures details of the contact surface between the bovine nail and the RHE of the 3D model, where fungal growth occurred. As a result, we were able to visualize the extent of fungal growth on the internal contact surface, even though the term “surface” used here is more generic, referring to what is visible in the image rather than the orientation of the nail in the 3D model. Given this, the impact of fungal growth and product applications on the nail structure can also be evaluated.

The forth paragraph of the Discussion should be written more concisely. It should also be preceded by a better transition, e.g., “In addition to the model’s ability to reflect effects of therapeutic agents 2 on fungal growth and invasion, it also captures potential side effects of topical treatments. For instance, several signs of compromised nail structure were evident after…”.

Suggestions of the reviewer are considered in the revised version of the manuscript. The fourth paragraph now reads:

In addition to the model’s ability to reflect effects of antifungal agents on fungal growth and invasion, it also captures potential side effects of topical treatments. For instance, several signs of compromised nail structure were evidenced after repeated applications of the product containing natural oils and monoterpenes. , SEM analyses revealed alterations in the lamellar structure of nail keratin, degradation of nail kera-tin fibers, and even the presence of holes after four days of daily application. Essential oils have been shown to induce a damaging effect on the skin stratum corneum struc-ture due to conformational changes of the lipid and keratin network in the stratum corneum [23], resulting in a change of the orderly and compact structure that increases the skin permeability and reduces the effect of barrier function. While the lipid and keratin content are different between skin stratum corneum and nail structures [24], our observation of the delamination of the keratin network in the bovine nail aligns with the previous report indicating a damaging effect on skin stratum corneum struc-ture and also with the use of  monoterpenes, especially linalool, as highly effective chemical penetration enhancers for the transungual delivery of antifungal agents [25]. Thus, considering the need for long-term treatment duration to achieve mycological cure, repeated use of monoterpene-based products may seriously harm the nail struc-ture and hinder normal nail regrowth. In addition, as terpenes such as limonene, lin-alool and geraniol are known flagrance allergens [26], an increased risk of developing allergies may be observed in case of repeated accidental applications of such products on the skin surrounding the nail. Therefore, products that are safe to use and do not impede nail regrowth should be prioritized when selecting appropriate treatment op-tions for individuals with onychomycosis. In contrast, repeated applications of Urgo Damaged Nails help preserve the lamellar keratin structure of the nail, similar to that of a healthy, non-colonized nail. No structural damage is observed suggesting that this product may be suitable for long-term repeated applications and effective in protecting the nail during the regrowth process.”

The Discussion should include a summary of limitations, e.g., with regards to the model’s unsuitability for long-term studies, lack of validation with additional fungal species, potential differences between bovine and human nails etc.

A summary of the limitations has been added to the discussion section of the revised manuscript. Although we could not find any literature on differences in composition and structure between human nails and bovine nail sheets, we acknowledge this as a potential limitation. The following paragraph has been added to the discussion:

Model limitations such as its unsuitability for long-term studies, lack of validation with additional fungal species, and potential differences between bovine and human nails should be considered when interpretating these results. Due to the high sensitivi-ty of in vitro 3D model with human tissue, an increased number of applications or longer treatment duration were not considered to ensure the viability of the system. The 3D model was colonized with T. rubrum the most representative trichophyton strain of onychomycosis. As different colonization capacities of Trychophyton strains were evidenced in vitro on bovine membranes [27], it would be interesting to evaluate the ability of other trichophyton such as Trichophyton mentagrophytes to growth, invade the 3D model and respond to antifungal agent.

Reference 27. Is added in the revised manuscript: Temporiti, M.E.E.; Guerini, M.; Baiguera, R.M.; Buratti, S.; Desiderio, A.; Goppa, L.; Perugini, P.; Savino, E. Efficacy of Bovine Nail Membranes as In Vitro Model for Onychomycosis Infected by Trichophyton Species. J. Fungi 2022, 8, 1133

The manuscript is well written but still requires thorough editing by a native English speaker due to several syntax and grammar issues (e.g., use of articles) and some unusual/non-scientific phraseology (e.g., “thanks to” in section 2.6).

The revised manuscript has been proofread by all the authors and is now written in clear and fluent English.

Round 2

Reviewer 3 Report

The authors have been highly responsive to my and the other reviewers' prior comments and have substantially improved the manuscript.

My only major remaining comment is that the authors should make it abundantly clear in the Introduction (in their scope/objectives statement) that this proof-of-concept study was designed to test "medical devices" and not "traditional antifungal agents". Otherwise, the average medical mycologist reading this article may be confused about this notable omission, especially when considering that "antifungal therapy" and "antifungal agents" were mentioned several times throughout the manuscript.

There are still some minor language issues, including in the newly added/edited sections; however, those should be addressable during editorial editing.

There also seem to be an issue with Figure 1, which didn't include a visual legend or any texts within the assembled manuscript file. This should be checked to ensure that all elements are properly displayed. The figure should be self-explanatory and shouldn't require such an extensive figure legend. I neither liked the original version of Figure 1 (too complicated and distracting) nor the current one (not self-explanatory, does very little to visualize the procedures).

Author Response

My only major remaining comment is that the authors should make it abundantly clear in the Introduction (in their scope/objectives statement) that this proof-of-concept study was designed to test "medical devices" and not "traditional antifungal agents". Otherwise, the average medical mycologist reading this article may be confused about this notable omission, especially when considering that "antifungal therapy" and "antifungal agents" were mentioned several times throughout the manuscript.

As suggested by the reviewer, we have clarified this point in the following parts of the manuscript:

- Line 6 of the abstract, “classified as medical devices” is now added. It reads “Four different products classified as medical devices were applied once daily on the nails: Urgo Damaged Nails (UDN), Excilor (EXlor), Poderm Purifying (PDermP), Poderm Booster (PDermB).

-Introduction, the last sentence is rewritten as follows:

“This proof-of-concept study was designed to evaluate the suitability of such a 3D model as a preclinical onychomycosis model and to test the effects of different topical medical devices and not traditional antifungal agents on fungal growth and nail structure.”

There also seem to be an issue with Figure 1, which didn't include a visual legend or any texts within the assembled manuscript file. This should be checked to ensure that all elements are properly displayed. The figure should be self-explanatory and shouldn't require such an extensive figure legend. I neither liked the original version of Figure 1 (too complicated and distracting) nor the current one (not self-explanatory, does very little to visualize the procedures).  

We’re proposing a new version of the Figure 1 and hope it will satisfy the reviewer expectations. We're now adding  visual legends to the figure to explain the presence RHE and difference between non-colonized and colonized RHE with T. rubrum, the covering with a bovine nail, the daily application of the product and finally the GMS and SEM analyses.

The legend text is thus simplified:

Figure 1. Set-up of the 3D model. a: non-colonized RHE ; b: colonized RHE with no product application; c: colonized RHE with product applications on the bovine nail. Blue arrow: colonization with T. rubrum. Grey oval: Bovine nail sheets covering of the RHE. Green arrow: once daily application of the product”.
